# On the Parametrization of Epidemiologic Models—Lessons from Modelling COVID-19 Epidemic

**DOI:** 10.3390/v14071468

**Published:** 2022-07-02

**Authors:** Yuri Kheifetz, Holger Kirsten, Markus Scholz

**Affiliations:** Institute for Medical Informatics, Statistics and Epidemiology, University of Leipzig, Haertelstrasse 16-18, 04107 Leipzig, Germany; holger.kirsten@imise.uni-leipzig.de

**Keywords:** COVID-19 epidemiologic models, parametrization, extended multi-compartment SIR-type model, input-output non-linear dynamical system, Bayesian knowledge synthesis

## Abstract

Numerous prediction models of SARS-CoV-2 pandemic were proposed in the past. Unknown parameters of these models are often estimated based on observational data. However, lag in case-reporting, changing testing policy or incompleteness of data lead to biased estimates. Moreover, parametrization is time-dependent due to changing age-structures, emerging virus variants, non-pharmaceutical interventions, and vaccination programs. To cover these aspects, we propose a principled approach to parametrize a SIR-type epidemiologic model by embedding it as a hidden layer into an input-output non-linear dynamical system (IO-NLDS). Observable data are coupled to hidden states of the model by appropriate data models considering possible biases of the data. This includes data issues such as known delays or biases in reporting. We estimate model parameters including their time-dependence by a Bayesian knowledge synthesis process considering parameter ranges derived from external studies as prior information. We applied this approach on a specific SIR-type model and data of Germany and Saxony demonstrating good prediction performances. Our approach can estimate and compare the relative effectiveness of non-pharmaceutical interventions and provide scenarios of the future course of the epidemic under specified conditions. It can be translated to other data sets, i.e., other countries and other SIR-type models.

## 1. Introduction

Predicting the spread of an infectious disease is a pressing need as demonstrated for the present SARS-CoV-2 pandemic. Due to the worldwide high disease burden, a plethora of mathematical epidemiologic models was proposed. This includes auto-regressive time series methods, Bayesian techniques, and application of deep learning methods, but also mechanistic models and hybrid models combining some of these approaches (see [1] for a review). Among mechanistic models, based either on agents [2,3,4] or on compartments [5], the most commonly proposed and published model type is the classical SIR (S = susceptible, I = infected, R = recovered) type compartment model, which was presented with different modifications often considering further aspects and details such as disease states, age structure, contact patterns, and intervention effects. Major aims of these models are to predict (1) the dynamics of infected subjects, (2) requirements of medical resources during the course of the epidemic, or (3) the effectiveness of non-pharmaceutical intervention programs (NPI) [6,7]. Examples for models addressing these aims are a SECIR model (E = exposed, C = cases) proposed by Khailaie et al. [8], a SEIR type model proposed by Barbarossa et al. [9], models from The Robert-Koch institute [10], and from Dehning et al. [11]. 

A good prediction performance does not only depend on the precise structure of the model but on its parametrization. This, however, is a non-trivial and often underestimated task due to the following issues applicable to other infectious diseases as well: key epidemiologic parameters are often unknown or only known within ranges. Therefore, parametrization based on observational data is a common approach. However, reported official data bases are heterogeneous and often biased due to (1) lag in reporting of cases/events [12], (2) changing testing policy either due to limited testing capacities, which might depend on the pandemic situation itself or by changing risk profiles of people to be tested (e.g., defined risk groups, dependence on symptoms, degree of prophylactic testing) [13], and (3) incompleteness of data [14]. Moreover, parametrization depends on further epidemiologic issues to be considered, comprising (1) changing age-structure of the infected population with impact on symptomatology, hospital or intensive care requirements and mortality, (2) spatial heterogeneity of the spread of the disease driven by local conditions and outbreaks, (3) new pathogen variants becoming prevalent, (4) non-pharmaceutical interventions continuously updated in response to the pandemic situation, and finally, (5) the progress of vaccination programs and its effectiveness.

Due to these interacting complexities, it is close to impossible to construct a fully mechanistic model covering all these aspects in parallel. Therefore, we here propose a framework of epidemiologic model parametrization, which accounts for these issues in a more phenomenological, data-driven way applicable even for limited or biased data resources.

In detail, we here propose to integrate a mechanistic epidemiologic model as a hidden layer into an input-output non-linear dynamical system (IO-NLDS), i.e., the true epidemiologic dynamics cannot be directly observed. This allows distinguishing between features explicitly modelled (in our case different virus variants, vaccination) and changing factors of the system which are difficult to model mechanistically (in our case changes of contacts, e.g., due to non-pharmaceutical interventions or changing contact behavior, changing age-structure of the infected population and changes in testing policy, in the following abbreviated as NPI/behavior). These factors are imposed as external inputs of the system.

We then estimate parameters by a knowledge synthesis process considering prior information of parameter ranges derived from different external studies and other available data resources such as public data. We are thus going beyond previous modeling approaches that only used point estimates for parameters [15,16]. Specifically, we use Bayesian inference for the parameter estimation, which could also be time-dependent. We analyze the structure of available public data in detail and translate it to model outputs linked by an appropriate data model to the hidden states of the IO-NLDS, i.e., the epidemiologic model. We demonstrate this approach on an example epidemiologic model of SECIR type for SARS-CoV-2 and data of Germany and Saxony, but our method can be translated to other countries, other models or even other infectious disease contexts.

## 2. Materials and Methods

### 2.1. General Approach

We consider input-output non-linear dynamical systems (IO-NLDS) originally proposed as time-discrete alternatives to physiological pharmacokinetic and –dynamics differential equations models [17]. This class of models couples a set of input parameters such as external influences and factors with a set of output parameters, i.e., observations by a hidden model structure to be learned (named *core model* in the following). This coupling is not necessarily fully deterministic, i.e., data are not required to represent directly state parameters of the model. This represents a major feature of our approach in order to account for different types of biases in available observational time series data.

We here demonstrate our approach by using an epidemiological model as core of the IO-NLDS. Non-pharmaceutical interventions, changes of testing policy, age distributions and severity of disease courses were phenomenologically modelled by external control parameters imposed on the epidemiologic model via the input layer of the IO-NLDS. Random influxes of infected subjects e.g., by travelling activities or outbreaks are also considered by this approach. Number of reported infections, intensive care (IC) cases, and deaths are considered as output parameters not directly representing the hidden states of the model due to several data issues including reporting delays. The model is then fitted to data by a full information approach, i.e., all data points were evaluated by a suitable likelihood function.

The single steps of this process are explained in detail below.


**Assumptions of the core model**


We adapted a standard SECIR model (SECIR = susceptible, exposed, cases, infectious, recovered) for pandemic spread. We introduced an asymptomatic compartment in order to account for infected patients, which do not have symptoms, a common condition of SARS-CoV-2 infection. A compartment of patients requiring intensive care (IC) was added to model respective requirements and we distinguished between deceased and recovered patients.

We subdivided most of the compartments into three sub-compartments with first order transitions to model time delays. Transition rates between sub-compartments are the same for each respective compartments for the sake of parsimony. This approach is extensively used in pharmacological models [10]; it is equivalent to a Gamma-distributed transit time [11]. To allow for two concurrent virus variants with differing properties, compartments of asymptomatic and symptomatic infected subjects are duplicated. This allows us, for example, to simulate the take-over of the more infectious B.1.1.7 (Alpha), and later, B1.617.2 (Delta) variant observed e.g., in all European countries [12].

The general scheme of the IO-NLDS system is shown in Figure 1. We make the following assumptions:The input layer consists of external modifiers influencing (1) reporting policy (e.g., changing testing policy), (2) rates of infections (affected by non-pharmaceutical interventions, age structure, influx of cases), and (3) risks of severe disease conditions such as IC requirements and deaths, also depending on the changing age structure of infected subjects.The output layer of observable data is linked to the hidden layers of the core model by specific data models (see later).Susceptible, non-infected people (Sc): We assume that 100% of the population is susceptible to infection at the beginning of the epidemic.The latent state E comprises infected but non-infectious people.The asymptomatic infected state IA has three sub-compartments (I_(A,1), I_(A,2) and I_(A,3)). From I_(A,1), transitions to the symptomatic state or the second asymptomatic state are possible. From I_(A,2), only transitions to I_(A,3) and then to the recovered state R are assumed.The symptomatic infected state IS is also divided into three compartments (I_(S,1), I_(S,2), and I_(S,3)). The sub-compartment I_(S,1) comprises an efflux toward the sub-compartment C_1 representing deteriorations toward critical disease states. Otherwise, the patient transits to I_(S,2). From I_(S,2), a patient can either die representing deaths without prior intensive care or transit to I_(S,3). Finally, the efflux of I_(S,3) flows into R representing resolved disease courses.Both cases I_A and I_S contribute to new infections but with different rates to account for differences in infectivity and quarantine probabilities.The compartment C represents critical disease states requiring intensive care. We assume that these patients are not infectious due to isolation. Again, the compartment is divided into three sub-compartments, C_1, C_2, and C_3. In C_1, a patient can either die or transit to C_2, C_3, and finally, R.Patients on the recovered stage R are assumed to be immune against re-infections.We duplicate the compartments E, I_(A,1),…, I_(A,3), I_(S,1),…, I_(S,3) to account for two concurrent virus variants. We assume different infectivities for the two variants. All other parameters are assumed equal. No co-infections are assumed.

These assumptions are translated into a difference equation system (see Appendix A). Model compartments and their properties are explained in Table 1.

All model parameters of the model are described in Table 2. Complete dynamics of the epidemic in Germany is shown in Figure 2 and Figure 3.

### 2.2. Input Layer

The input layer represents external factors acting at the SECIR model, effectively changing its parameters [42]. We define step functions *b*_1_ and *b*_2_ as time-dependent input parameters modifying the rate of infections caused by asymptomatic, respectively symptomatic subjects. To identify dates of change, we used a data-driven approach based on a Bayesian Information Criterion informed by changes in non-pharmaceutical interventions for Germany based on Government decisions, changing testing policies as well as events with impact on epidemiological dynamics such as holidays or sudden outbreaks. Details can be found in Appendix B.

We also accounted for changes in the probabilities of critical courses and mortality, which can be explained by changes in testing policies covering asymptomatic cases to a different extent (for example symptomatic testing only vs. introduction of screening tests, e.g., rapid antigen tests), respectively shifts in the age-distribution of patients or changes in patient care efficacy (new pharmaceutical treatment, overstretched medical resources). Again, this is implemented by step functions *p_crit_*, respectively *p_death_*. Number of steps are determined on the basis of a Bayesian Information Criterion. Details can be found in the Appendix B as well as in Table A5 from Appendix I and Table A8 from Appendix J. The parameter *P_S,M_* represents the percentage of reported infected symptomatic subjects in relation to all symptomatic subjects. This value is assumed to be constant (50%) in the present version of the model. We describe the parameters defining the input layer in Table 3.

### 2.3. Output Layer and Data

We fit our model to time series data of reported numbers of infections *I_S,M_*, deaths *D_M_*, and occupation of ICU beds *C_M_* representing the output layer of our IO-NLDS model. Data source of infections and deaths were official reports of the Robert-Koch-Institute (RKI) in between 4 March 2020 and 29 March 2021. Number of critical cases were retrieved from the German Interdisciplinary Association of Intensive and Emergency Medicine (Deutsche Interdisziplinäre Vereinigung für Intensiv- und Notfallmedizin e.V.—DIVI) in between 25 March 2020 and 29 March 2021. Time points in proximity to Christmas and the turn of the year (19 December 2020 to 19 January 2021) were heavily biased and therefore omitted during parameter fitting.

However, also for the considered time intervals several sources of bias need to be considered. We handled these issues as explained in the following:

*Infected cases:* We first smoothed reported numbers of infections with a sliding window of seven days centered on the time point of interest to control for strong weekly periodicity. We assume that these numbers correspond to a certain percentage *P_S,M_* of symptomatic patients. This is justified by the fact that the majority of reported infected cases develop symptoms (about 85% according to the RKI [43]), but there is also a large amount of asymptomatic cases (approximately 55–85% of infections [37,38,39]. In the present model, we assume *P_S,M_* as constant. The exact equation linking states of the SECIR model with the measured numbers of infected subjects *I_S,M_* can be found in Appendix B and Appendix C Equation (A7). We further accounted for delays in the reporting of case numbers by introducing a log-normally distributed delay time as explained in Appendix C.

*Critical cases:* The number of critical COVID-19 cases (DIVI reported ICU) is available since end of March 2020 [44]. We assumed that these data are complete since 16 April 2020 when reporting became mandatory by law in Germany. Earlier data were up-scaled from the number of reporting hospitals to the number of ICU-beds of all hospitals according to the reported ICU capacity available for 2018. We coupled the sum of the critical sub-compartments *C_i_* (*i* = 1,2,3) to these numbers directly.

*Deaths:* Deaths are reported by the RKI but daily reports do not reflect true death dates, which needs to be accounted for. Available daily death data of the RKI are retrospectively updated with a delay between true death date and reported date (death report delay—DRD). We assume that the DRD is normally distributed with an average of 7.14 days and a standard deviation of 4 days as reported by Delgado et al. [45]. Details can be found in Appendix C.

*Occurrence of B.1.1.7 variant:* In January 2021, the variant B.1.1.7 became endemic in Germany and quickly replaced all other variants. Onset of this development was modeled by an instantaneous influx of 5% of newly infected subjects into the *E^Mu^* compartment on 26 January 2021 estimated from published data [46].

### 2.4. Parametrization

We carefully searched the literature to establish ranges for our model parameters. These ranges are used as prior constraints during parametrization of our model (Table 2). Justification of prior values is provided in Appendix H. Parameters are then derived by fitting the predictions of the model to reported data of infected subjects, ICU occupation, and deaths using the link functions of model and data explained in the previous section. This is achieved via likelihood optimization. Likelihood is constructed using the same principles as reported [47]. In short, the likelihood consists of three major parts, namely the likelihood of deviations from prior values, the likelihood of residual deviations from the data, and a penalty term to ensure that model parameters are within the prescribed ranges, as explained in Appendix D. We follow a full-information approach intended to use all data collected during the epidemic as explained in Appendix E, Appendix F and Appendix G. As a result, our model fits well to the complete dynamics of the epidemic in Germany in the above mentioned time period (Figure 2 and Figure 3).

To ensure identifiability of parameters, we checked a number of parsimony assumptions. For example, we assumed that the dynamical infection intensities of asymptomatic ($b_{1}\cdot r_{1}\left(t\right)$) and symptomatic subjects ($b_{2}\cdot r_{2}\left(t\right)$) are proportional with factor $rb_{1,2}$. We also determined Bayesian Information criteria (BIC) for different partitioning numbers of the external jump functions ($N_{crit}$ and $N_{death}$) to keep these as small as possible. Details can be found in Appendix B.

Likelihood optimization is achieved using a stochastic approximation of an estimation-maximization algorithm (SAEM) [48]. The algorithm is based on a stochastic integration of marginal probabilities without using likelihood approximations such as linearization or quadrature approximation or sigma-point filtering [17].

Confidence intervals of model predictions are derived by Markov-Chain-Monte-Carlo simulations, i.e., alternative parameter settings were sampled from the parameter space around the optimal solution (Appendix B, Appendix F and Appendix G). We use these parameter sets to simulate alternative epidemic dynamics. This resulted in a distribution of model predictions from which empirical confidence intervals are derived.

### 2.5. Implementation

The model and respective parameter estimations are implemented in the statistical software package R from which external publicly available functions are called. The model’s equation solver is implemented as C++ routine and called from R code using the Rcpp package. The code and data for simulation of the output layers using the reported parameter settings will be made available via our Leipzig Health Atlas: (https://www.health-atlas.de/models/40, accessed on 26 June 2022) and GitHub (https://github.com/GenStatLeipzig/LeipzigIMISE-SECIR, accessed on 26 June 2022) [49].

## 3. Results

### 3.1. Explanation of Epidemiologic Dynamics

We used the full data set to explain the course of infections, ICU occupations, and deaths between 4 March 2020 and 29 March 2021 in Germany. A total of three parameters were assumed time-dependently, namely Infectivity *b*_1_ and the probability of a critical disease course (*p_crit_*) and death (*p_death_*). We identified nine fixed and 19 empirically identified time points of NPI/behavioral changes (Table A1 from Appendix I). Regarding *p_crit_* and *p_death_*, we identified 18 respectively 19 time steps (See Table A2 and Table A3 from Appendix I and Table A7 from Appendix J). Throughout the epidemic, we observed a good agreement of our model and incident (Figure 2) and cumulative data (Figure 3). Corresponding residual errors are provided for all observables (Table A4 from Appendix I). As shown in Table A2 from Appendix I, we estimated 14 static and three dynamically changing parameters using 1170 data points (390 daily measurements of registered cases, registered deaths and ICU occupancy) for Saxony as well as for Germany.
Figure 2Agreement of model and incident data. We show incident infections, deaths, and daily ICU occupancy during the course of the epidemic in Germany in between 4 March 2020 and 29 March 2021. Comparison of IO-NLDS model (magenta curve) and data (thin grey curves = raw data, solid black curve = data averaged by sliding window) is provided in the upper column. A good agreement is observed (shaded area = prediction uncertainty, vertical lines = changes in NPI/contact behavior). The middle row represents the corresponding input layer, i.e., the estimated time course of the time-dependent input parameters, namely infectivity and probabilities of critical disease course and death. Time steps correspond to the lines of changing NPI/contact behavior as displayed in the upper row. In the lower row, we present percentages of B.1.1.7 among infected subjects (first figure), subjects older than 80 years among infected corresponding to high death tolls (second), and subjects in the age categories 35–59, respectively 60–79 among critical cases (last figure of last row).
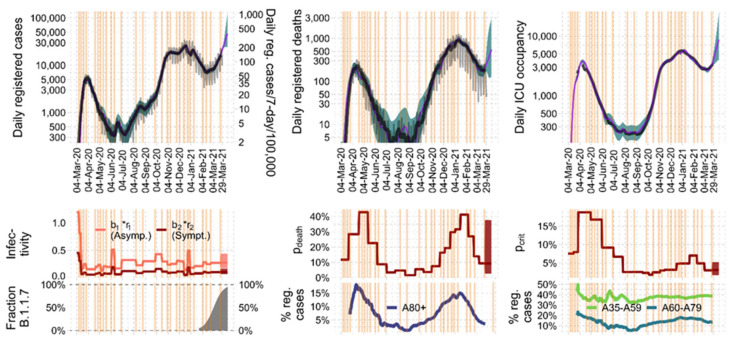

Figure 3Agreement of model and cumulative data. We show cumulative infections and deaths during the course of the epidemic in Germany in between 4 March 2020 and 29 March 2021. Comparison of IO-NLDS model (magenta curve) and data (solid black curve) is provided. A good agreement is observed (shaded area = prediction uncertainty, vertical lines = changes in NPIs/contact behavior).
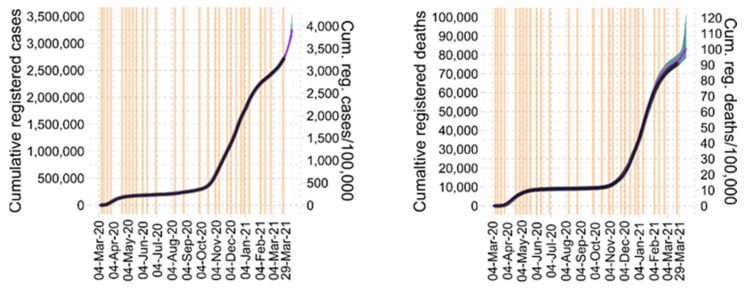


### 3.2. Parameter Estimates and Identifiability

Parameter estimates of the SECIR model are presented in Table 2, while those required to define the input layer are presented in Table 3 and Table A1 from Appendix I. For those parameters for which we used prior information for fitting purposes, we compared the respective expected posteriors with their best priors (see Figure 4). Statistics are provided in Table A5 from Appendix I. No significant deviations between expected values of posteriors and priors were detected. All relative errors of parameters of the SECIR model are smaller than 10% demonstrating excellent identifiability of all epidemiologic parameters. As expected, identifiability of the external control functions is reduced. Largest standard errors of steps are in the order of 70% still demonstrating reasonable identifiability (Table A3 from Appendix I).

### 3.3. Plausibilization of Estimated Step Functions of Infectivity

We estimated the infectivity as an empirical step function through the course of the epidemic. This step function should also roughly reflect NPI effectivity. We therefore compared our infectivity step function with the Governmental stringency index of NPI as estimated on the basis of Hale et al. [50]. Results are displayed at Figure 5 and revealed a reasonable agreement.

### 3.4. Model Predictions

We regularly used our model to make predictions regarding the future course of the epidemic. Predictions were specifically made for the Free State of Saxony, a federal state of Germany and were published at the Leipzig Health Atlas [49]. We here present comparisons of our predictions with the actual course of the epidemic for two scenarios to demonstrate utility of our approach. Parameter values for Saxony were obtained in the same way as for Germany restricting available data of infected subjects, ICU cases, and deaths to this state. Estimated parameter values are presented in Table A6, Table A7 and Table A8 from Appendix J.

While Saxony was almost spared from the first wave of SARS-CoV-2 in Germany, the second wave hit the country particularly hard resulting in the highest relative death toll of all German states (1 out of 400 inhabitants of Saxony died from COVID-19 during the second and the immediately following B.1.1.7-driven third wave). The second wave was on its peak in the middle of December 2020. A hard lock-down was initiated at this time including closure of schools, prohibition of all team-based leisure activities, and night-time curfew. We were asked by the government to estimate the length of lock-down required to break the second wave. Stringency of lock-down was comparable to the first wave. Thus, we simulated four scenarios: an optimistic assumption of a lock-down efficacy of 60% reduction in infectivity, a more realistic scenario with 40% reduction, a pessimistic assumption of only 20% lock-down efficacy, and finally, 0% reduction (no lock-down) as control scenario. Results are shown in Figure 6 and revealed a good agreement of our prediction with the actual course for the 40% scenario considered likely.

At the beginning of February 2021, the second wave was broken in Saxony and first relaxations of NPIs were conducted. At this time, the more virulent B.1.1.7 variant became endemic in Germany. At 14 February, the true percentage of the B.1.1.7 variant was unknown due to lack of sequencing capacities. Moreover, there were uncertainties with respect to the increase in infectivity by the B.1.1.7 variant. We therefore simulated three scenarios (optimistic: 10% initial proportion of B.1.1.7, infectivity increased by factor 1.7, expected: 20% initial proportion, 1.8-times increase in infectivity, pessimistic: 30% initial proportion, 2-times increase in infectivity). Results are shown in Figure 7. The actual course of the epidemic was close to the pessimistic scenario, i.e., the second wave was directly followed by a third wave due to the B.1.1.7 variant. Indeed, later data revealed that the proportion of B.1.1.7 was already close to 30% (pessimistic assumption) at the time the simulation was performed. Moreover, our model correctly predicted the variant replacement by B.1.1.7.

## 4. Discussion

In this paper, we propose a method of parametrization of COVID-19 epidemiologic models and applied it to an extended SECIR-type model to explain the course of the epidemic in Germany and one of its federal state, the Free State of Saxony. Moreover, we demonstrated how the model can be used to make relevant predictions, which could be validated on the basis of subsequent observational data.

A key idea of our approach is the embedding of differential equations-based epidemic modelling into an input-output dynamical system (IO-NLDs). This has two major advantages. First, the approach allows combining explicit mechanistic models of epidemic spread and phenomenological considerations of external impacts on model parameters via the input layer. This allows parametrizing models of different complexity. For example, in our model we non-explicitly considered the effect of age structures of the diseased population by time-dependent input parameters such as probabilities of critical disease courses and deaths. This could easily be replaced for example by age-structured models. We believe that such a combined empirical/mechanistic approach is well suitable to address the complexity of COVID-19 epidemic dynamics for which it is impossible to consider all relevant mechanisms explicitly and in parallel.

The second major advantage of our approach is that we assumed a non-direct link between state parameters of the embedded SECIR model and observables. This allows interposing a data model considering known biases of the available data resources. We aimed at identifying relevant bias sources as far as possible and considered them in our proposed data models. However, these data models could be subjected to changes in the future for example if better data of COVID-19-related death will be released. Improved data models could be easily integrated into our framework.

Note that the IO-NLDS implementation translates the embedded differential equations model to a discrete scale (i.e., days in our case), which however appears to be sufficient for describing an epidemic.

We also want to note that the SECIR model used here is by far neither unique nor the most comprehensive one. For example, The Robert-Koch institute developed a model for the purpose of estimating the effect of different vaccination strategies which could easily be included into our SECIR-type models [54]. Although integration of differential equations-based models into our IO-NLDS context is more straightforward, our approach is also applicable to agent-based models. In general, the aspect of parameter estimation of such models is underdeveloped in view of the highly biased data resources used and to our knowledge, no generic concept was proposed so far.

Based on our IO-NLDS formulation and data models, we parametrized our model on the basis of data of infection numbers, critical cases, and deaths available for Germany and Saxony. Here, we chose a full-information approach considering all data in between start of the epidemic 4 March 2020 to 29 March 2021. We also applied a Bayesian learning process by considering other studies to inform model parameter’s settings. Thus, we combine mechanistic model assumptions with results from other studies and observational data. This approach is very popular in pharmacology [55] but despite its importance it is yet rarely applied in epidemiology [11]. It resulted in a complex likelihood function, which is optimized on the basis of Markov-Chain Monte Carlo (MCMC) algorithms, as we described in Appendix B and Appendix F. If the likelihood has a unique maximum, most of the samples eventually accumulate in its vicinity after a certain number of “burn-in” steps. This allows an effective MCMC search of the best parameter estimates as well as approximations of their standard errors (standard deviations of the sample) and the degree of overfitting. However, if parameters are interdependent, MCMC algorithm samples manifolds of alternative solutions, resulting in large standard errors of the overfitted parameters. We successfully addressed this issue by a modified version of Maire’s algorithm [56]. Central to this approach is the idea that the proposal distribution adapts to the target by locally adding a mixture component when the discrepancy between the proposal mixture and the target is deemed to be too large. In other words, this algorithm samples multidimensional parameters sets, approximating it as a mixture of multivariate Gaussian distribution. Such approaches enable adequate sampling of model parameters and detection of overfitting as well as of multiple local maxima of the likelihood. Our results revealed small standard errors indicating lack of overfitting, see Table A1 and Table A3 from Appendix I, Table A6 and Table A7 from Appendix J. We also applied rigorous information criteria to limit the number of steps of our input functions. As a consequence, it was possible to identify both, the fixed parameters of the SECIR model and the time-variable input functions representing changing NPI/contact behavior and age-structures.

Model parametrization resulted in a good and unbiased fit of data for the period considered for Germany. Fixed parameter values of the SECIR model did not significantly deviated from their prior values if available. It required 18 respectively 19 steps of changes of the probabilities to develop critical stage and to die respectively. A total of 13 intensification and 15 relaxation events were necessary to describe the epidemic dynamics over the time course of observations. Estimated infectivity roughly correlated with the Governmental Stringency Index [51]. We regularly contributed forecasts of our model to the German forecast Hub [57].

We also demonstrated utility of our model by several mid-term simulations of scenarios of epidemic development in Saxony, a federal state of Germany. We could show that predictions of reported infections were in the range of later observations for scenarios considered likely.

As future extensions and improvements of our model, we will consider stochastic effects on a daily scale, for example to model random influxes of cases or to model random extinctions of infection chains. These effects are relevant to be considered in times of low incidence numbers such as those observed in Germany in the summers 2020 and 2021. Our IO-NLDS framework is well suited to implement such extensions [17].

In future versions of our model, we will also include age-structures and implement a vaccination and waning model in analogy to other research groups. In the current version of the model, we assumed a constant proportion of symptomatic patients reported as infected. This does not consider for example changing testing policies (i.e., symptomatic vs. prophylactic testing). We plan to refine our model in this regard in the future. Finally, we will consider the Delta and Omicron variants emerging in 2021 [53] in the next update of our SECIR model.

In summary, the primary focus of the paper is an adequate parametrization of epidemiological models on the basis of complex, possibly biased data, as well as its coupling with structurally unknown dynamical external influences. This approach allows for a clear separation of mechanistic model compartments from random or time-dependent non-mechanistic influences and biases in the data. We believe that this approach is useful not only for the parametrization of the SECIR model presented here but also for other epidemiologic models including other disease contexts and data structures.

## Figures and Tables

**Figure 1 viruses-14-01468-f001:**
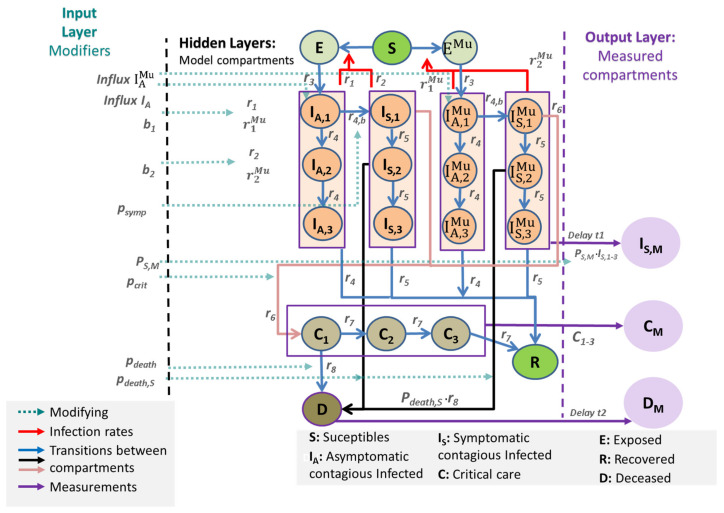
General scheme of our IO-NLDS model. The epidemiologic SECIR model is integrated as a hidden layer. Respective equations are provided in Appendix A. The input layer consists of external modifiers including parameter changes due to changes in testing policy, non-pharamceutical interventions, and age-structures. The output layer is derived from respective hidden layers via stochastic relationships (see later). The output layer is compared with real-world data. The superscript Mu denotes new virus variants.

**Figure 4 viruses-14-01468-f004:**
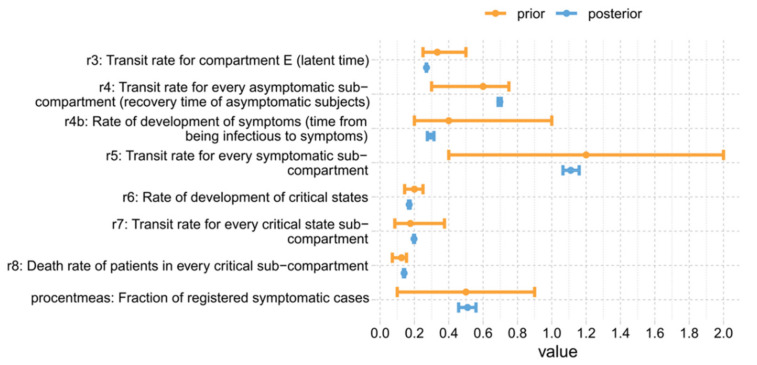
Comparison of prior and posterior values of estimated parameters of the SECIR model. We present prior vs. posterior distributions of estimated parameters of the SECIR model. Ranges for priors represent assumed minimum and maximum values. Ranges for posteriors represent 95%-confidence intervals. Numbers are provided in Table A8 from Appendix J.

**Figure 5 viruses-14-01468-f005:**
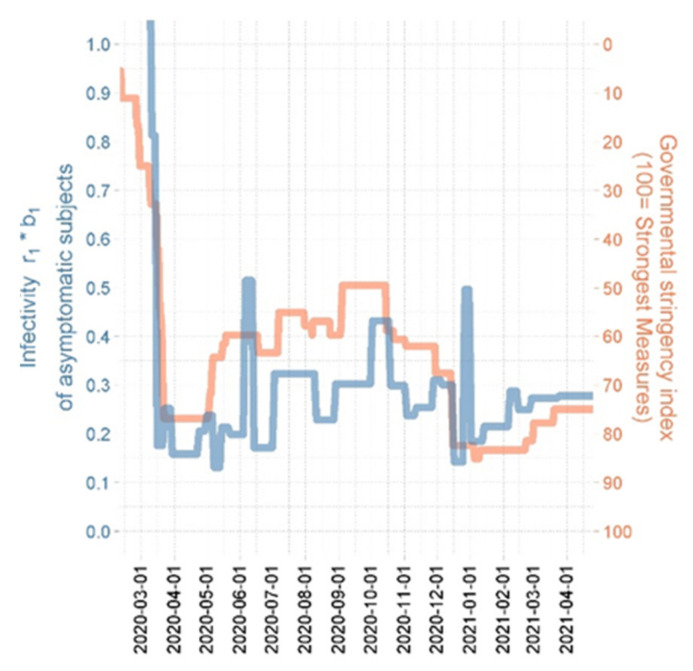
Relationship between estimated step function of infectivity of asymptomatic subjects and the Federal Government stringency index (GSI). The GSI [50] is a composite measure based on nine response indicators including school closures, workplace closures, and travel bans, rescaled to a value from 0 to 100 (100 = strictest). If policies vary at the level of federal states, the index is shown for the state with the strictest measures. For background info see also [51]. Colors of curves correspond to different y-axes.

**Figure 6 viruses-14-01468-f006:**
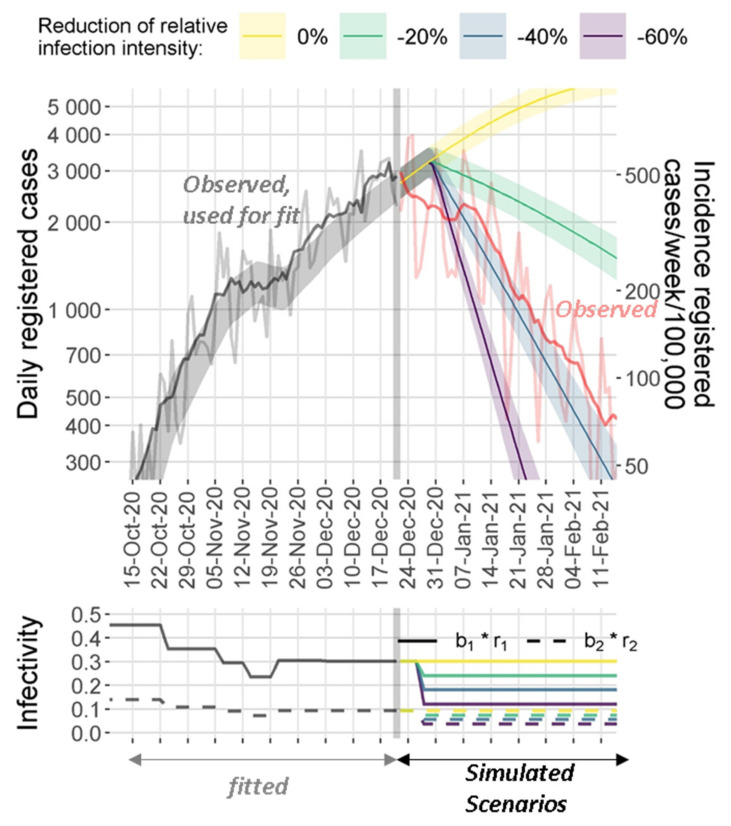
Comparison of predicted and observed decline of the second wave in Saxony according to the initiated lock-down. Our model was used to fit the observed data until 21 December 2020 (shown as grey curve (raw data) and black curve (smoothed) of reported test-positives). Estimated step functions b_1_ and b_2_ describing the infectivity of asymptomatic and symptomatic subjects were reduced by 0% (yellow: no lock-down = control scenario), 20% (green: pessimistic scenario), 40% (blue: realistic scenario), and 60% (magenta: optimistic scenario) to simulate four scenarios of the future course of the epidemic under lock-down conditions. The observed numbers of test-positives after the 21 December 2020 are shown in red (light red = raw data, dark red = smoothed) closely followed the expected scenario of 40% lock-down efficacy. Shaded areas represent 95% prediction intervals. The predictions and parameters were reported in our regular bulletin deposited at Leipzig Health Atlas, ID: 85AH9JMUFM-4.

**Figure 7 viruses-14-01468-f007:**
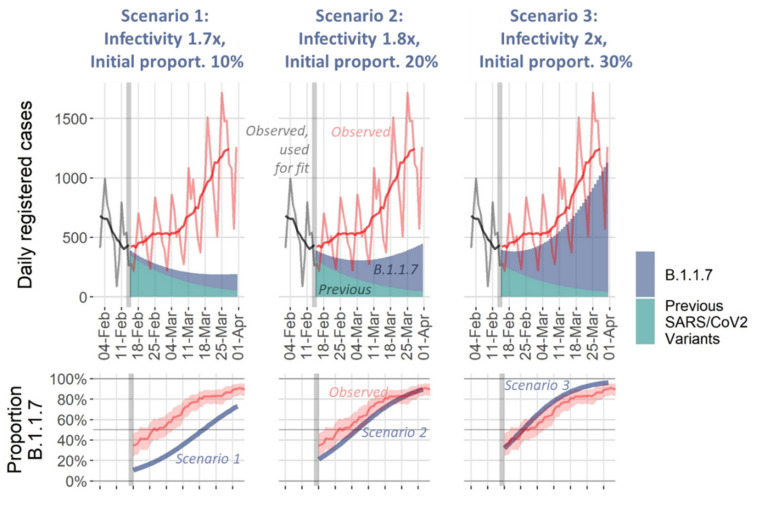
Simulation of third wave scenarios for Saxony/Germany: Upper row: The model was used to fit all observed data until 14 February 2021 (grey curve = raw data of reported testpositives, black curve = smoothed). Three scenarios were simulated differing in assumed initial proportion of B.1.1.7 which was not exactly known at this time point (10%, 20%, and 30%, respectively) and in the assumptions regarding increased virulence of B.1.1.7 (parameter mur = 1.7, 1.8 and 2, respectively). Predicted course of subjects infected with the respective variants are shown as shaded areas. The observed total numbers of testpositives (light red = raw data of reported testpositives, dark red = smoothed curve) closely followed the pessimistic scenario 3. Lower row: When comparing the proportion of B.1.1.7 as retrieved from [52] from 18 July 2021, initial proportion of B.1.1.7 was indeed close to that assumed for scenario 3. Blue curves represent 95% confidence intervals of the B.1.1.7 proportion predicted for the different scenarios. All predictions were reported in our bulletin at the 20 February 2021 deposited at our Leipzig Health Atlas [53].

**Table 1 viruses-14-01468-t001:** Description of model compartments. We describe the compartments of the model and their biological meaning. Compartments E, $I_{A}$, and $I_{S}$ are duplicated to account for two concurrent virus variants.

Compartment Name	Sub-Compartments	Description
Sc		Susceptible
*E*		Latent stage (not infectious)
$I_{A}$	$I_{A,1}$	Asymptomatic infected state 1, can either develop symptoms, i.e., transit to $I_{S,1}$ with probability $p_{symp}$ and rate *r*_4*b*_ or stays asymptomatic with probability $1-p_{symp}$ and transits to $I_{A,2}$ with rate *r*_4_
	$I_{A,2}$	Asymptomatic infected state 2, transits to $I_{A,3}$ with rate *r*_4_
	$I_{A,3}$	Asymptomatic infected stage 3 transits to *R* with rate *r*_4_
$I_{S}$	$I_{S,1}$	Symptomatic infected state 1. Can either become critical, i.e., transits to $C_{1}$ with probability $p_{crit}$ and rate *r*_6_ or stays sub-critical with probability $1-p_{crit}$ and transits to $I_{S,2}$ with rate *r*_5_
	$I_{S,2}$	Symptomatic infected state 2, can either die, i.e., transits to *D* with probability $p_{death,S}$ or transits to $I_{S,3}$ with probability $\left(1-p_{death,S}\right)$ and rate *r*_5_
	$I_{S,3}$	Symptomatic infected state 3, transits to *R* with rate *r*_5_
*C*	$C_{1}$	Critical state 1, not infectious. Can either die, i.e., transits to *D* with probability $p_{death}$ and transit rate *r*_8_ or stays critical with probability $1-p_{death}$ and transits to $C_{2}$ with rate *r*_7_
	$C_{2}$	Critical state 2, transits to *C*_3_ with rate *r*_7_
	$C_{3}$	Critical state 3, transits to *R* with rate *r*_7_
*R*		Recovered (absorbing state)
*D*		Dead (absorbing state)

**Table 2 viruses-14-01468-t002:** Basic model parameters. We present prior values and ranges derived from the literature as well as estimated values derived from parameter fitting. Transit rate means reverse of transit time of the respective compartment. Posteriors can be found in Figure 4. §: Further details and definitions on parameters are given in the Appendix A Equations (A1) and (A2), where also a justification of priors is provided, (Appendix H).

Parameter	Unit	Description	Source	Reference	Value	Prior Mean	Min	Max
influx	Subjects per day	Initial influx of infections into compartment *E* until first interventions	Estimated	§	3171	-	-	-
$r_{1}$	Day^−1^	Infection rate through asymptomatic subjects	Estimated	§	1.19	-	-	-
$r_{2}$	Day^−1^	Infection rate through symptomatic subjects	$\mathrm{Set}\;\mathrm{equal}\;\mathrm{to}\;rb_{1,2}\cdot r_{1}$ (parsimony)	§	0.451	-	-	-
$rb_{1,2}$	-	Proportion of infection rate symptomatics/asymptomatics *r*_1_*/r*_2_	Estimated	§	0.379	-	0	-
$r_{3}$	Day^−1^	Transit rate for compartment *E* (latent time)	prior constraint	§, [10,18,19,20,21]	0.272	1/3	1/4	1/2
$r_{4}$	Day^−1^	Transit rate for asymptomatic sub-compartments	prior constraint	§,[22,23,24,25]	0.636	3/5	3/10	3/4
$r_{4,b}$	Day^−1^	Rate of development of symptoms after infection	prior constraint	§, [10,18,19,20,21,26,27,28]	0.456	2/55	1/5	1
$r_{5}$	Day^−1^	Transit rate for symptomatic sub-compartments	prior constraint	§	0.946	6/5	6/15	6/3
$r_{6}$	Day^−1^	Rate of development of critical state after being symptomatic	prior constraint	§, [10,29,30,31]	0.186	1/5	1/7	1/4
$r_{7}$	Day^−1^	Transit rate for critical state sub-compartment	prior constraint	§,[10,32,33,34]	0.159	3/17	3/35	3/8
$r_{8}$	Day^−1^	Death rate of patients in critical sub-compartment 1	prior constraint	§, [29,35,36]	0.104	1/8	1/14	2/13
$p_{symp}$	-	Probability of symptoms development after being infected	Set or prior constrained (overfitted if estimated unconstrained)	§,[37,38,39]	0.5	-	0.3	0.8
$p_{crit}$ $\left(p_{crit,0}\right)$	-	Initial value $p_{crit,0}$ of step function $p_{crit}$, the probability of becoming critical after developing symptoms	Estimated	§, [9,27]	0.0765	-	0	1
$p_{death}$ ($p_{death,0}$)	-	Initial value $p_{death,0}$ of step function $p_{crit}$, the probability of dying after becoming critical	Estimated	§, [32]	0.119	-	0	1
$p_{death,S}$	-	Probability of death after developing symptoms without becoming critical	Set equal to $p_{death,S,0}\;\cdot p_{death}$ (parsimony)	§, [32]	-	-	0	1
$p_{death,S,0}$		Proportionality factor for probability of death after developing symptoms without becoming critical	Estimated	§	0.587			
$P_{S,M}$	-	Fraction of unreported cases	prior constraint	§, [40,41]	0.499	0.5	0.1	0.90
mur		Ratio of r1^Mu^/r1 = r2^Mu^/r2 reflecting higher infectivity of B.1.1.7 variant	Set	§	1.65	-	-	-

**Table 3 viruses-14-01468-t003:** Parameters used to define the input layer. These parameters were used to empirically model changing NPIs or changing contact behavior, changes in testing policies and changing age-structures during the course of the epidemic. Respective input functions constitute the input layer of our IO-NLDS model.

Parameter	Unit	Description	Source	Remarks
$N_{tr}$	-	Number of time points of changes of NPI/contact behavior	Empirically defined	13 intensifications, 15 relaxations identified (determined by information criterion)
$b_{tr,j}$, *j =* 1,…,$\;N_{tr}$	-	Relative infectivity of subjects in the time interval [tr, tr + 1]	Estimated	assumed to be the same for symptomatic and asymptomatic patients
$Tr_{j}$, *j* = 1,…,$\;N_{tr}$	Days	Time points of NPI/contact behavior changes	Estimated or fixed	Strictly monotone sequence
$N_{crit}$	-	Number of time steps of $p_{crit}\left(t\right)$	Empirically defined	18 (determined by information criterion)
$\alpha_{crit,j}$, *j =* 1,…,$\;N_{crit}$	-	Value of $p_{crit}$ between two time steps	Estimated	Within the interval [0, 1]
$T_{pcrit,j}$, *j =* 1,…,$\;N_{crit}$	Days	Time points of changes of $p_{crit}$	Estimated	Strictly monotone sequence
$N_{death}$	-	Number of time steps of $p_{death}\left(t\right)$	Empirically defined	19 (determined by information criterion)
$\alpha_{death,j}$, *j =* 1,…,$\;N_{death}$	-	Value of $p_{death}$ between two time steps	Estimated	Within the interval [0, 1]
$T_{pdeath,j}$, *j =* 1,…,$\;N_{death}$	Days	Time points of changes of $p_{death}\left(t\right)$	Estimated	Strictly monotone sequence
$Del_{tr}$	Days	Delay of activation of NPI	Fixed	2 days

## Data Availability

Code and data are shared. https://www.health-atlas.de/models/40 and GitHub https://github.com/GenStatLeipzig/LeipzigIMISE-SECIR.

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
