# Peer review of "On the Parametrization of Epidemiologic Models—Lessons from Modelling COVID-19 Epidemic"

_viruses, 2022, doi:10.3390/v14071468_

Round 1

Reviewer 1 Report

The study has addressed many experiments but the presentation is not well designed. The authors should carefully check the paper before submission as it is not professional; for example:

- Page 14, section 4 is discussion and the conclusion has merged with this section and followed by Acknowledgement (section 5) and page 16, again, discussion (section 4) and conclusion (section 5)!! which are borrowed from the template. As a team of three authors, none of them have checked them.

- The introduction is very short with lack of suitable literature review and should be improved extensively. There many mathematical models such as SIR, SEIR, etc. It is strongly recommended to rewrite the introduction and give more details regarding the developed models by the other researchers.

Author Response

Comment 1: The study has addressed many experiments but the presentation is not well designed. The authors should carefully check the paper before submission as it is not professional; for example:

- Page 14, section 4 is discussion and the conclusion has merged with this section and followed by Acknowledgement (section 5) and page 16, again, discussion (section 4) and conclusion (section 5)!! which are borrowed from the template. As a team of three authors, none of them have checked them.

Authors reply: We thank the reviewer very much for correction and apologize for this technical error. The unnecessary sections are removed.

Changes in manuscript: The unnecessary sections are removed.

Comment 2: -- The introduction is very short with lack of suitable literature review and should be improved extensively. There many mathematical models such as SIR, SEIR, etc. It is strongly recommended to rewrite the introduction and give more details regarding the developed models by the other researchers.

Authors reply: We thank the reviewer for pointing to this issue. We now provide a broader introduction mentioning other models and model classes used by other researchers. We also reorganized a section of the discussion, where we previously discussed and compared other models, now presenting them in the introduction as suggested.

We hope that our overview is now sufficiently comprehensive. In view of the large literature, we selected example publications of modelling classes. If the reviewer misses specific references or approaches, we are happy to include them.

Changes in manuscript:

We added respective paragraphs in the introduction and reorganized the discussion.

Reviewer 2 Report

The authors develop a prediction model for the number of infected (symptomatic and asymptomatic), critical states (1,2 and 3), recovered and dead people for the COVID-19 epidemic. The model is a combination of a SIR and non-linear dynamical system models as previously considered in [40]. The estimation of the parameters is done from a Bayesian perspective by the consideration of MCMC estimation algorithms.

My comments are mainly related to the statistical procedures of the proposed estimation methodology:

1)Appendix D: It is not clear if the authors are considering the likelihood or log-likelihood function (Equation S4.1 page 22, line 645). If it is the likelihood function it is not clear how to assign different weights to each of the three terms in Equation S4.1 in order to implement the estimation algorithms. In this same respect, it is important to be consistent in the use of lower case (ll) and lower case (LL) terms.

2)Since the proposed model is very complex, it is important to present evidence that the MCMC algorithms converge to the posterior distribution of the parameters. It is necessary to include some plot of the ergodic averages of the MCMC realizations to be sure that, with the number of iterations considered, we have some confidence that the MCMC has already converged to the stationary distribution that corresponds to the posterior distribution of the parameters.

3)The predictions are very good (without considering the seasonal effects) but it could be an effect that the number of parameters of the model is very large. Please, include some information on the total number of parameters and the total number of observations of the examples in the paper.

4)There are many references in the text that appear as "Error! Reference Source"

Author Response

Comment 1: Appendix D: It is not clear if the authors are considering the likelihood or log-likelihood function (Equation S4.1 page 22, line 645). If it is the likelihood function it is not clear how to assign different weights to each of the three terms in Equation S4.1 in order to implement the estimation algorithms. In this same respect, it is important to be consistent in the use of lower case (ll) and lower case (LL) terms.

Authors reply: We thank the reviewer for this important comment. We corrected all corresponding incorrect of ambiguous terminologies. In equation (S4.1), all weights are 1. The weights appear only inside of the term   (formula S4.5) and their values are given in the text (lines 684-690).

Changes in manuscript:

We corrected the following

  • likelihood to “negative log-likelihood” everywhere, where “negative” lacks (line 638, 639, 656, 674, 681, 695, 703, 709, 713, 730, 768)
  • nll to nLL (line 642)

Comment 2: Since the proposed model is very complex, it is important to present evidence that the MCMC algorithms converge to the posterior distribution of the parameters. It is necessary to include some plot of the ergodic averages of the MCMC realizations to be sure that, with the number of iterations considered, we have some confidence that the MCMC has already converged to the stationary distribution that corresponds to the posterior distribution of the parameters.

Authors reply: We thank the reviewer very much for this suggestion and we agree to provide Geweke convergence diagnostics for our Markov chains.

Changes in manuscript: Results of Geweke convergence diagnostic for our Markov chains are inserted in lines 826-849 (appendix F), including figure A.2 showing that a stationary sampling distribution is achieved.

Comment 3: The predictions are very good (without considering the seasonal effects) but it could be an effect that the number of parameters of the model is very large. Please, include some information on the total number of parameters and the total number of observations of the examples in the paper.

Authors reply: We thank the reviewer for this important remark and we provide this  information of the manuscript.

Changes in manuscript: We added lines 288-291, informing that about 1,170 data points are available and that 14 static and three dynamically changing input parameters were fitted for Saxony as well as for Germany.

Comment 4: There are many references in the text that appear as "Error! Reference Source"

Authors reply: We are sorry for this technical issue. However, we could not detect it in our submitted version; neither in the word document nor the pdf version generated by the online submission system (see our first submission http://tiny.cc/proof_Kheifetz_Viruses for the proofs generated by the system). We could only assume that this is an issue of the submission platform. We describe this issue in the cover letter asking the editor for help in this regard.

Round 2

Reviewer 1 Report

I am happy with the revision.